# *Syzygium aromaticum* Extracts as a Potential Antibacterial Inhibitors against Clinical Isolates of *Acinetobacter baumannii*: An In-Silico-Supported In-Vitro Study

**DOI:** 10.3390/antibiotics10091062

**Published:** 2021-09-01

**Authors:** Abdelhamed Mahmoud, Magdy M. Afifi, Fareed El Shenawy, Wesam Salem, Basem H. Elesawy

**Affiliations:** 1Department of Botany and Microbiology, Faculty of Science, Al-Azhar University, Assuit 71524, Egypt; abdelhamedmahmoud35@yahoo.com (A.M.); magdybadran.136@azhar.edu.eg (M.M.A.); FareedElshenawy.42@azhar.edu.eg (F.E.S.); 2Department of Botany and Microbiology, Faculty of Science, South Valley University, Qena 83523, Egypt; 3Department of Pathology, College of Medicine, Taif University, P.O. Box 11099, Taif 21944, Saudi Arabia

**Keywords:** antibiotic-resistant genes, docking, GC-Mass, imipenem, urine samples, penicillin-binding proteins, virulence genes, wound swab

## Abstract

Imipenem is the most efficient antibiotic against *Acinetobacter baumannii* infection, but new research has shown that the organism has also developed resistance to this agent. *A. baumannii* isolates from a total of 110 clinical samples were identified by multiplex PCR. The antibacterial activity of *Syzygium aromaticum* multiple extracts was assessed following the GC-Mass spectra analysis. The molecular docking study was performed to investigate the binding mode of interactions of guanosine (Ethanolic extract compound) against Penicillin- binding proteins 1 and 3 of *A. baumannii*. Ten isolates of *A. baumannii* were confirmed to carry *rec*A and *iut*A genes. Isolates were multidrug-resistant containing *bla*_TEM_ and *Bla*_SHV_. The concentrations (0.04 to 0.125 mg mL^−1^) of *S. aromaticum* ethanolic extract were very promising against *A. baumannii* isolates. Even though imipenem (0.02 mg mL^−1^) individually showed a great bactericidal efficacy against all isolates, the in-silico study of guanosine, apioline, eugenol, and elemicin showed acceptable fitting to the binding site of the *A. baumannii* PBP1 and/or PBP3 with highest binding energy for guanosine between −7.1 and −8.1 kcal/mol respectively. Moreover, it formed π-stacked interactions with the residue ARG76 at 4.14 and 5.6, Å respectively. These findings might support the in vitro study and show a substantial increase in binding affinity and enhanced physicochemical characteristics compared to imipenem.

## 1. Introduction

One of the principal global health hazards is recognized by the Multi-Drug Resistant (MDR) gram-negative bacterial diseases which contribute worldwide to nosocomial infections [1]. In Egypt, antimicrobial resistance monitoring could be used as a prerequisite to avoid nosocomial infections [2]. *Acinetobacter baumannii* is a gram-negative aerobic coccobacillus, common in hospital settings, particularly in intensive care (ICU). Septicemia, pneumonia, endocarditis, meningitis, and dermatitis are among the bacterial nosocomial infections they cause [3]. *A. baumannii* is generally antibiotic-resistant because of decreased permeability, efflux pumping systems, inactivation of enzymes, and the formation of biofilm [4,5]. Thus, they are often β-lactams, aminoglycosides, and quinolones resistant [6]. Several virulence factors such as colicin V production (cvaC), curli fibers (csg), siderophores such as aerobactin (iutA), and cytotoxic necrotizing factor (cnf) are responsible for the pathogenicity of *A. baumannii* [7]. β-lactam antibiotics such as cephalosporins, carbapenems, and penicillins represent approximately 60% of the used antibiotics [8]. *A. baumannii* resistance is mainly due to Extended-Spectrum Beta-Lactamases (ESBLs); which could disrupt various β-lactam antimicrobial agents as penicillins and their derivatives [9]. ESBLs are encoded by specific genes such as; bla*_TEM_* (encoded for penicillins-resistance), bla*_SHV_* (sulfhydryl variable), and bla*_CTX_* (encoded for cephalosporins-resistance) [10,11,12]. Many researchers have identified the causes of this resistance to pathogens. One of the mechanisms relating to beta-lactam bacterial resistance is the reduction of binding affinity between penicillin-binding proteins (PBP) and beta-lactam antibiotics [13,14]. The earliest in vitro experimentations have demonstrated that imipenem and/or sulbactam binds to penicillin-binding proteins (PBPs) on *Acinetobacter* spp. accordingly, it has been proposed that bacteria are killed by this mechanism. [15,16]. Detection of latent virulence genes and/or antibiotic resistance genes in clinical isolates of *A. baumannii* has several epidemiologically important findings that allow the research to track the spread of this bacterium’s infectious diseases [17]. To combat multi-drug resistant gram-negative bacteria and their biofilm, natural phytocompounds that mimic enzymes are necessary. [18,19]. The discovery of new products used for the treatment of severe diseases was reported in herbal medicines as a valuable source [20]. Many plant species like *Syzygium aromaticum* have different pharmacological and antibacterial actions because of their constituents, such as glycosides, hormones, tannins, alkaloids, and saponins [21,22]. Cloves (*Syzygium aromaticum*, syn. *Eugenia aromaticum*, or *Eugenia caryophyllata*) are the aromatic desiccated buds [23]. It belongs to the genus Eugenia (family Myrtaceae). Clove is known universally. It is used mostly in food, medicinal products, perfume, and cosmetics. [24]. There is also a compelling requirement to identify novel pharmacological targets and to understand the function of possible therapies in the treatment of MDR *A. baumannii* infections. In the present study, the prevalence, antibiogram, PCR detection of virulence, and antibiotics resistant genes of *A. baumannii* isolated from different clinical samples were investigated. Subsequently, the phytochemical investigations of *Syzygium aromaticum* different extracts were carried out using gas chromatography-mass spectrometer technique (GC-MS) (Thermo scientific^™^ Technologies Australia, Trace^™^ 1310 Series). The antibacterial activity of the extracts or imipenem was tested in vitro against *A. baumannii* isolates. Docking studies were conducted to determine and compare the interactions of both *Syzygium aromaticum* compounds and the antibiotic imipenem inside the bacterial outer membrane enzymes’ active sites.

## 2. Results

### 2.1. Patients and Clinical Characteristics

A total of 110 patients with clinical evidence of nosocomial infection (respiratory diseases, elevated liver and/or kidney functions, etc.) were enrolled in the study (Table 1). Of them, 63 (56.88) were females and 47 (43.12%) were males. The majority (50.9%—38 female; 18 male) of participants were found in the age group between 30 and 60 years. Thirty patients (7 female; 23 male) with a percentage of approximately (27.3%) of the study participants were above 60 years old, while 21.8% (18 female; 6 male) were below 30 years old. Table 1 depicts the clinical characteristics of the study participants.

#### The Incidence of *A. baumannii* Isolates among Examined Clinical Samples

The prevalence of the isolated bacteria in the clinical samples is illustrated in Figure 1. Among the 110 clinical specimens, the highest percentage of *A. baumannii* was 66.04% followed by 18.87% and 15.09% was recorded for sputum, wound, and urine samples respectively (Figure 1). The total bacterial count in urine culture showed that 70% (21/30) of female patients had a range of 10^4^ to 10^5^ CFU/mL compared to 65% (13/20) of male patients. While 5% of female and 2% of male patients had CFU values ≥ 10^6^.

### 2.2. Antimicrobial Susceptibility Testing

A total of 10 (9.09%) of *A. baumannii* isolates were recovered from sputum samples (*n* = 7; 70%), urine (*n* = 2; 20%) and pus (*n* = 1; 10%). The rate of resistant isolates to a panel of antibiotics with different potency was illustrated as MIC values (µg mL^−1^) in a heat- map as shown in Figure 2. All isolates (*n* = 10; 100%) displayed a high resistance pattern to Ticarcillin, Ticarcillin/Clavulanic Acid, Piperacillin, Piperacillin/Tazobactam, Cefotaxime, Cefepime, Ciprofloxacin, and Amikacin. Around (80%) are resistant to Gentamicin and Tobramycin; and (70%) are resistant to imipenem and meropenem. An intermediate resistant levels with (60 and 40%) to Trimethoprim/Sulfamethoxazole and Minocycline respectively. Interestingly, sensitivity levels within Colistin and Minocycline were observed. According to the Vitek-2 compact system test, three out of 10 *A. baumannii* isolates (codes A4, A7, and A9) were found to share the same phenotype and antibiotic pattern profile. Hence, these isolates were skipped. The rest of the isolates that showed different phenotypes and antibiotic patterns such as (A1, A2, A3, A5, A6, and A8) were considered for further experiments.

#### Detection of Virulence and Antibiotic-Resistant Genes

The multiplex PCR screening for virulence and antibiotic-resistant genes showed that 100% of *A. baumannii* were carrying the rec*A* and *iut*A virulence genes (Table 2; Figure 3). On the other hand, the antibiotic-resistant genes, *bla*_TEM_ was present in all *A. baumannii* isolates. *Bla*_SHV_ was present with a percentage of approximately 83% since the isolate (A8) missed that gene (Table 2; Figure 3).

### 2.3. Gas Chromatography-Mass Spectrometry (GC/MS) Analysis

Plant content analysis and extraction play an important role in the progress, restoration, and quality management of herbal formulations. As a result, one of the primary goals of this analysis was to identify the bioactive compounds found in the *S. aromaticum* extracts to assess their role in improving the antibacterial activity against *A. baumannii* isolates. Eighteen signal peaks related to separate components were obtained by gas chromatography-mass spectrometry (GC/MS) in the aqueous extract of *S. aromaticum* (Table 3). The main constituents were α-pinene (18.82%); Beta-caryophyllene (15.12%); oleic acid (14.52%); camphor (11.75%); globuolol (11.35%); Loganetin (8.51%); Apioline (5.45%) and Hexadecanoic acid (4.59%). The ethanol extract possesses 12 signal peaks for compounds such as oleic acid (27.22%); Guanosine (8.91%); indole (6.83%) and 1-Eicosene (6.3%). Finally, the highest percentage content in the ethyl acetate extract gives Linoleic acid (36.16%); Citral (13.48%), and Hexadecanoic acid (11.95%). Other active compounds with their peak number, concentration (peak area %), and retention time (RT) are presented in Table 3.

### 2.4. In Vitro Assay for the Antibacterial Activity of S. aromaticum Extracts

Antibacterial activity of *S. aromaticum* different extracts and Imipenem against *A. baumannii* isolates was analyzed by minimal inhibitory concentrations (MIC) by determining the bacterial viability using a colorimetric INT-formazan assay. As a result, we determined the minimal bactericidal concentrations (MBC) which confirmed the killing of *A. baumannii* isolates over time (24 h). The individual use of *S. aromaticum* aqueous and/or ethyl acetate extracts against *A. baumannii* isolates (A1, A2, A3, A5, A6, and A8) exhibited MBC values varying from 0.17 to 0.25 mg mL^−1^, respectively (Figure 4A,C). Within isolates A1 and A6, ethanol extracts revealed MBC with significant values (0.04 to 0.125 mg mL^−1^) respectively when compared to Imipenem. (Figure 4B). It is worth mentioning that Imipenem showed a great bactericidal efficacy against all isolates with a concentration of 0.02 mg mL^−1^ are adequate to kill all the tested *A. baumannii* isolates (Figure 4).

### 2.5. Molecular Docking Studies of Standard Antibiotic and Herbal Ligands

From the obtained data in Table 4, the antibiotic (Imipenem) suited the binding sites of (PBP1 and PBP3) well, with binding energies ranging from −6.8 to −6.5 kcal/mol respectively (Figure 5). Moreover, the molecular docking simulations were performed for Guanosine, Apioline, Eugenol, and Elemicin against the target proteins (PBP1 and 3) to support the in vitro study via their mode of interactions (Figure 6 and Figure 7). The screened compound (guanosine) displayed respectable fitting to the same binding sites of the targets and having binding energies of −7.1 to −8.1 kcal/mol, respectively. The compound docked to the target protein PBP1 through HB interactions with the residues GLN285 and TYR415 at 2.98 and 2.03 Å, respectively. In addition, it formed two types of interactions such as HB and π- stacking with the target protein PBP3. The guanosine exhibited HBs with the residues ARG71, ARG76 and TYR192 at the distances of 2.97, 2.84, 2.93, and 2.25 Å, respectively. Moreover, it formed π-stacked interactions with the residue ARG76 at 4.14 and 5.6 Å, respectively (Figure 5).

The extracted phyto-compounds (Apioline, Eugenol, and Elemicin) were successfully docked to PBP1 with the binding energies −5.6, −5.4 and −5.3 Å, respectively (Table 4, Figure 6). Apioline docked with the target through π-sigma interaction with the residue TYR707 at 3.60 Å. While, Eugenol docked through π-cation interaction with ARG298 at 4.70 Å. Finally, Elemicin formed two π-cation interactions with the residue ARG236 at 5.48 and 4.12 Å, respectively. For the second enzyme PBP3, the molecules exhibited binding affinities to the active site pockets with docking scores −6.0, −5.8, and −5.2 Å, respectively. They showed π-stacking, similar to π-sigma and π-cation interactions with the residues TYR450 and Lys339 (Table 4, Figure 7).

## 3. Discussion

The risk of antibiotic-resistant nosocomial infections becomes life-threatening in the intensive care unit and other areas of hospital care [25]. Multi-drug resistant bacteria such as *A. baumannii* and *P. aeruginosa* cause many of these infections [6]. The current study showed the proportion of nosocomial infections due to the gram-negative bacilli *A. baumannii* isolated from clinical samples as urine (15.09%); wound (18.87%); and sputum (66.04%) among the patients (Figure 1). These bacteria are a common cause of urinary tract infections (UTIs) that upset the kidney, leading to pyelonephritis, as well as the bladder, resulting in cystitis [26]. UTI symptoms included elevated kidney function and high levels of *A. baumannii* in the patients’ sputum (66.04%) (Table 1). This result is conceivable since UTI symptoms are not a reliable indication of illness. The presence of bacteria must be confirmed by urine culture in order to diagnose UTI [27]. The urine culture in our results indicated the presence of bacteriuria in 70% of female patients and 65% of male patients. The percentage of *A. baumannii* in our study was in agreement with the results obtained by Al-Agamy et al. [10] the highest prevalence of *A. baumannii* isolates was in respiratory samples followed by wound samples and urine (Figure 1). Our findings are in agreement with the results obtained by Abdulzahra et al. [28]. The antibiotic susceptibility test for *A. baumannii* isolates indicated that all isolates (*n* = 10; 100%) were multi-drug resistant to Ticarcillin, Ticarcillin/Clavulanic Acid, Piperacillin, Piperacillin/Tazobactam, Cefotaxime, Cefepime, Ciprofloxacin, and Amikacin, while 70–80% are extensively drug-resistant. Resistance decreased to 40% with Minocycline and all isolates were sensitive to Colistin (Figure 2). Our findings are in covenant with earlier results concerning the MDR isolates [29]. Some of the most significant virulence genes of *A. baumannii* are colicin V production, curi fibers (*csg*), siderophores, such as aerobactin (*iut*A), and cytotoxic necrotizing factors (*cnf*) [30,31]. Virulence genes such as *rec*A and *iut*A were present in all isolates of *A. baumannii* (Table 2; Figure 3). The same findings were confirmed earlier for 18.75% of *A. baumannii* isolates from the hospital environment were carrying *iut*A genes [32]. Antibiotic-resistant genes *bla*_TEM_ and *bla*_SHV_ were the most common in *A. baumannii* as (100%) and (83%) respectively (Table 2; Figure 3). This finding is consistent with the results reported by Beriş et al. [11,33], where the prevalence of *bla*_TEM_ and *bla*_SHV_ was 55.7% and 7.7%, respectively. The highest percentage of *bla*_TEM_ (100%) and absence of *bla*_CTX_ in all isolates are in concert with the findings reported by Al-Agamy et al. [10]. The current study confirmed the promising efficacy of all *Syzygium aromaticum* extracts against all *A. baumannii* isolates with the lowest MBC values varied from 0.04 to 0.125 mg mL^−1^ that recorded for the ethanolic one (Figure 4). This could be due to the presence of some active phytochemicals such as guanosine, α- Pinene, Beta-caryophyllene; oleic acid; camphor, globulol, and loganetin that were detected in the GC-MS analysis (Table 3). α-pinenes were detected earlier as the main component in *S. aromaticum* aqueous extract [34]. The bactericidal activities of α-pinenes against both gram-negative and positive bacteria were reported earlier by Mercier et al. [35]. α-pinenes kill bacteria by damaging membrane structure and function [36]. A membrane’s expansion and fluidity were enhanced as a result of its lipophilic nature. Beta-caryophyllene is the most common antibacterial constituent in essential oils of the *Syzygium* genus, it is detected in *S. aromaticum*, *S. cumini*, *S. polyanthum* and *S. samarangense* [37]. The mechanism of action was discussed recently as Beta-caryophyllene alters the bacterial membrane permeability and causes non-selective pore formation that kills the bacterial cells [38]. Oleic acid in *Syzgium aromaticum* could inhibit bacterial growth by inhibiting the bacterial fatty acid synthesis [39]. The antibacterial activity of camphor present in lavender essential oil was reported by De Azeredo et al. [40]. Another important phyto-component such as globulol was discussed for their antifungal and antibacterial activity [41]. Limonene has previously been reported in an aqueous extract of *Syzygium aromaticum* by Jimoh et al. [42]. Finally, guanosine has been reported as a potent antibacterial activity both in vitro and in vivo [43]. Moreover, in silico molecular docking and modeling techniques [44,45] were performed in the current study for the ethanolic extracted compound (guanosine) for better understanding their mode of action through the interaction with the active site pockets of PBP1 and PBP3, and to identify new inhibitors as antibacterial agents. The result showed that the target compounds exhibited binding energy higher than standard drug imipenem (−7.1 and −8.1 Å, respectively) against the target proteins (Table 4; Figure 5, Figure 6 and Figure 7). The SAR (structure activity relationship) analysis showed that the antibacterial activity of guanosine can be modulated by the presence of pyrimidine and indole moieties, and hydroxyl and amino groups (Figure 5). Therefore, guanosine may be considered a good inhibitor for PBP1 and PBP3 proteins. In recent times, computational modeling and molecular-dynamic simulations have shown that phytoligands, such as Apioline, Eugenol, and Elemicin, have good binding potential with minimum binding energy and that they are stabilized by an interaction between antibacterial imipenem (carbapenam) of the present three generations, with their common objectives in *A. baumannii* [14]. Virtual screening findings and drug-like tests have shown that phyto-compounds are a hit molecule due to a much greater binding affinity (−9.4 kcal/mol) compared with imipenem in their target PBP1 of *A. baumannii* [13]. Displayed guanosine as a relevant screened molecule that has binding energies of −7.1 to −8.1 kcal/mol at the binding site for PBP1 and PBP3 proteins, respectively. Form the tabulated data in Table 4; we can conclude that guanosine has the best docking score and intermolecular interactions as compared with the other selected compounds (Apioline, Eugenol, and Elemicin). As a result, it might be a promising pharmacological option for combating *A. baumannii*. Given the increasing importance of pyrimidine compounds, particularly in recent years, which have several applications in various fields, most notably in medicine as antibacterial agents [46]. In addition, pyrimidines are nitrogen heterocyclic aromatic compounds with great interest as they constitute an important class of natural products. The substituted pyrimidines such as Brodiprim, Iclaprim, Trimethoprim, and Pyrimethamine are biologically important compounds and act as effective antibacterial drugs [47]. Therefore, Guanosine with pyrimidine ring and other phyto-compounds such as Apioline, Eugenol, and Elemicin was selected as drug candidate against PBP1 and PBP3 proteins.

## 4. Materials and Methods

### 4.1. Sampling Collection, Isolation, and Processing

One hundred and ten specimens; urine (*n* = 50), sputum (*n* = 40), and wound swabs (*n* = 20) were randomly collected under complete aseptic conditions from different hospitals and clinics in Qena province, Egypt (from January to December 2018). In total, 5 mL of clean-catch urine from patients suspected of urinary tract infection (UTI) is obtained in a sterile container. Urine samples were inoculated on MacConkey agar and/or Blood agar (Merck, Germany) at 37 °C for 24 h and observed for bacterial growth. Blood agar colonies were counted using a colony counter and checked for significant bacteriuria. The culture that grew ≥10^5^ CFU/mL, was measured as significant bacteriuria. For heterogeneous colonies, sub-culturing of individual distinct colonies was performed to ensure pure cultures. The sterile cotton swabs dipped into normal saline with the Levine method were obtained aseptically from two wound samples of each participant [48]. Sputum samples were taken from patients over a self-induced cough into sterile cups and sent for culture. Samples were primarily identified using typical laboratory approaches including growth on MacConkey agar and/or Blood agar (Merck, Germany) and Gram staining. Plates were incubated at 37 °C and examined for detectable bacterial growth after 48 h of incubation. The isolated bacteria were further identified using the automated system Vitek-2 (bioMérieux, France).

### 4.2. Phenotypic Identification of the Isolates by Vitek-2 Systems

The bacterial isolates were identified with the Automated Identification Biomerieux Vitek-2 System via morphological, classical biochemical studies. In total, the 41 tests included 18 sugar assimilation and fermentation; 2 decarboxylase tests, and 3 different tests on the 64 ID-GNB plastic well (for urease, utilization of malonate, and tryptophane deaminase). With a vacuum card, the organism’s 0.50 McFarland suspension is inoculated, made from a blood agar plate of 18–20 h (BioMe’rieux), and is automatically screened and inserted manually inside the Vitek-2 reader. The fluorescence of the inoculator is measured every 15 min [49]. All isolates identified as *Acinetobacter baumannii* were used for further characterization.

### 4.3. Antimicrobial Susceptibility Testing

For all recovered isolates, antibiograms were calculated according to the disc diffusion method mentioned earlier and compared with the standard chart (CLSI, 2017) [50]. MIC was detected by Vitek-2 compact system (bioMérieux, France), the susceptibility of *Acinetobacter baumannii* isolates was tested for 15 antibiotics (Bioanalyse ^®^). The used antibiotics were Amikacin, Tobramycin, Gentamicin, Ticarcillin, Ticarcillin/Clavulanic Acid, Piperacillin, Piperacillin/Tazobactam, Cefotaxime, Cefepime, Imipenem, Meropenem, Ciprofloxacin, Levofloxacin, Trimethoprim/Sulfamethoxazole and Minocycline. Multidrug resistance was defined as resistance to three or more antibiotics of the different classes [51]. *A. baumannii* isolates with the same phenotype and antibiotic pattern profile were excluded, only different isolates were considered for further experiments.

### 4.4. Recognition of Virulence and Antibiotic-Resistant Genes of A. baumannii Isolates

Molecular representation of the recovered *A. baumannii* was approved by multiplex PCR. The detection of eight encoding genes of virulence and antibiotic resistance was performed by using 16 primers. Set extraction of DNA was carried according to QIAamp DNA mini kit instructions. Genes encoding different virulence factors *rec*A and *iut*A. For antibiotic resistance genes bla_TEM_, and bla_SHV_ were analyzed by multiplex PCR.

### 4.5. DNA Amplification for the Selected Virulence and Antibiotic Resistance Genes of Isolates

As previously stated, DNA extraction was carried out under QIAamp DNA mini-kit instructions (QIAGEN, Germany, GmbH) [52]. Concisely, 200 μL of the sample suspension was inoculated with 10 μL of proteinase K and 200 μL of lysis buffer at 56 °C for 10 min. The lysate was added to 200 μL of 100% ethanol following incubation. The sample was washed and centrifuged according to the recommendations of the manufacturer. Nucleic acid was eluted with 100 μL of elution buffer provided in the kit. PCR amplification was performed using oligonucleotide primer (METABION, Germany) that were utilized in a 25 μL reaction containing 12.5 μL of EMERALDAMP Max PCR Master Mix (TAKARA, Japan), 1 μL of each primer of 20 pmol concentration, 5.5 μL of dist. water and 6 μL of DNA template. Applied thermal cyclers have been used to react in the biosystem. Table 5 summarizes all amplicon sizes and cycling conditions. The products of PCR were separated by electrophoresis on 1.5% agarose gel (APPLICHEM, Germany, GmbH) in 1xTBE buffer at room temperature using gradients of 5 V/cm. For gel analysis, 15 μL of the products were loaded in each gel slot. Gelpilot 100 bp and 100 bp plus ladders (QIAGEN, Germany, GmbH) and GeneRuler 100 bp ladder (FERMENTAS, THERMO) was used as a marker for electrophoresis to determine the fragment sizes. The gel was photographed by a gel documentation system (ALPHA INNOTECH, BIOMETRA) and the data were analyzed through computer software (AUTOMATIC IMAGE CAPTURE, USA).

### 4.6. Plant Material and Extraction

*Syzygium aromaticum* (clove) was purchased from a local market in Qena city, Egypt. Dry seeds were washed with sterile water and further dried, ground into a fine powder using a tissue grinder (IKA^®®^ A10, Germany). Plant species have been visually imaged for documentation under the Department of Botany and Microbiology, Science Faculty, South Valley University, Qena, Egypt, for further taxonomic identification purposes. Three solvents were used for the extract of bioactive components from *Syzygium aromaticum*, as follows: 10 g of plant powder was soaked separately in 100 mLof hot distilled water, ethanol, and/or ethyl acetate for 7 days with continuous stirring (150 rpm) at room temperature by using a bigger bill shaker, USA. The obtained extracts were filtered through a Buchner funnel with Whatman No.1 filter paper and evaporated by a rotary evaporator (BUCHI R-114, Switzerland) under reduced pressure to dryness at 45 °C. All extract residue were dissolved in dimethyl sulfoxide (DMSO) except the aqueous extract, which dissolved in sterile distilled water at a concentration of 100 mg/mL [56]. All extracts were sterilized using a syringe filter equipped with a 45 μm membrane filter, then kept at −4 °C.

### 4.7. Determination of the Minimum Inhibitory Concentration (MIC) and Minimum Bactericidal Concentration (MBC) by INT Reduction Assay

The determination of MIC and MBC were assayed as described earlier [57]. The freshly prepared culture of *A. baumannii* isolates was adjusted to OD_595_ of 0.01. 100 μL of each isolate culture was put into sterilized 96-well plates. Then, 20 μL of the original *S. aromaticum* extracts (100 mg mL^−1^) was added (serial dilutions of 10^−1^–10^−10^ were used, eight replicates were made for each dilution into complete raw of the 96-well plate). Imipenem (10 mg mL^−1^) and un-inoculated media were tested as the positive and negative control, respectively. After 24 h incubation at 37 °C, MIC was determined by the addition of 40μL of *p*-iodonitrotetrazolium violet chloride (INT) (0.2 mg/mL, Sigma-Aldrich) to the plates and re-incubated at 37 °C for 30 min. The lowest concentration which banned color change is the MIC [58,59]. The MBC was determined by transferring 50 mL from each well of overnight MIC plates (and/or higher) to sterile (TSA) fresh plates. Viable colonies were counted after 24 h at 37 °C. The limit of detection for this assay was 10^1^ CFU/mL.

### 4.8. Gas Chromatography-Mass Spectrometry (GC-MS) Analysis

*S. aromaticum* different extracts were subjected to gas chromatography-mass spectrometer technique (GC-MS) (Thermo scientific^™^ technologies, Trace^™^ 1310) with capillary column TG-5 (30 m × 250 μm × 0.25 μm) system were used. The mass detector used in split mode and helium gas with a flow rate of 1.5 mL/min was used as a carrier. The injector was operated at 230 °C and the oven temperature for the initial setup was 60 °C for 2 min ramp 10/min to 300 °C for 8 min. Mass spectra were taken at 70 eV, total GC running time was 35 min.

### 4.9. Molecular Docking Studies of Standard Antibiotic and Herbal Ligands

The objective of this work is to create a new, sensitive, and possible imipenem ligand derivative against the target *A. baumannii* penicillin-binding protein 1 and/or 3 (PBP1–PBP3). It begins by studying the crystal structure of imipenem treated PBP1 and PBP3, finding the active site of the protein (pocket) and examining interactivities with imipenem. A further step is to create novel ligand molecules from *S. aromaticum* derived bioactive compounds, including guanosine based on the imipenem structure complex with PBP1/3; the calculations will then be completed to calculate the free binding energy and physical-chemical characteristics of every molecule. To identify the causes of shifting powers, a binding interaction between the chosen analogs and the receiver may also be analyzed. The 3D structures of *A. baumannii* PBP1 and PBP3 were downloaded from the protein data bank [60]. The PDB files were energy minimized and optimized by the removal of water molecules and atomic clashes to get a stable confirmation. A receptor grid was then generated at the centroid of the active site cavity to perform the screening approach [61,62]. The ligand molecule was sketched using Chem Draw Ultra 0.7, then converted to SDF format using Open Babel software [63]. The docking study was carried out using the PyRx-virtual screening tool [64].

### 4.10. Statistical Data Analysis

Data were analyzed using the Mann–Whitney U test or a Kruskal–Wallis test followed by post hoc Dunn’s multiple comparisons. Differences were considered significant at *p* values of ≤0.05. For all statistical analyses, GraphPad Prism version 8 was used.

## 5. Conclusions

*A. baumannii* represents vital problems because of the high percentage of antibiotics resistance, their encoding gene for virulence, and antibiotics resistance. Although ethanol extract of *Syzygium aromaticum* was promising for the in vitro study against *A. baumannii*, improving the potency of beta-lactam antibiotics can be an overwhelming pre-condition for antibiotic resistance; it is thus vital that new analogues of *A. baumannii* PBP1 and/or PBP3 imipenem are developed, with a substantially higher relative binding energy free. The results show how imipenem analogues may be designed using a phyto-compound such as guanosine, apioline, eugenol, and elemicin against the target proteins (*A. baumannii* PBP1 and/or PBP3) in the in-silico drug design. This stimulating analogue exhibits increased physico-chemical characteristics, as well as greater binding affinity. Therefore, guanosine and other biologically active compounds extracted from *S. aromaticum* are eco-friendly and might be employed as an alternative antimicrobial agent against *A. baumannii* isolates.

## Figures and Tables

**Figure 1 antibiotics-10-01062-f001:**
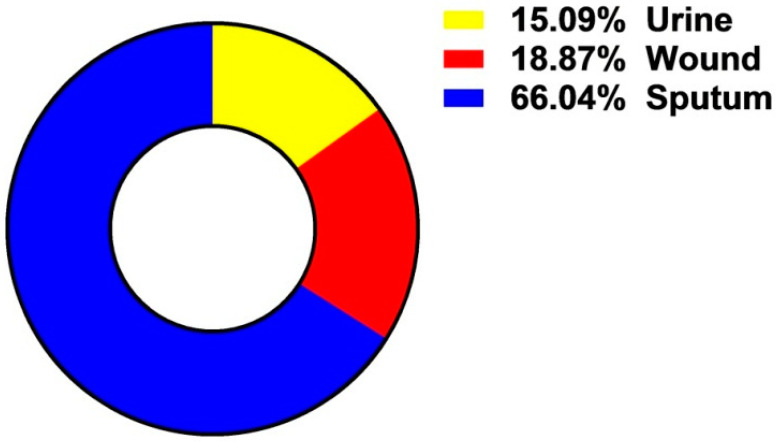
The incidence of *Acinetobacter baumannii* isolates among examined clinical samples.

**Figure 2 antibiotics-10-01062-f002:**
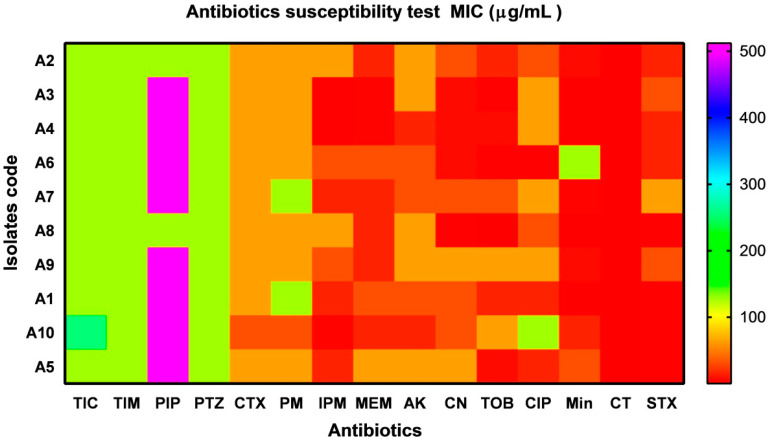
The heat-map illustrates the antibiograms for all the recovered *A. baumannii* isolates (A1: A10) by ViteK-2 systems. The intensity of colors indicates the numerical value of the MIC (µg/mL). TIC; Ticarcillin: TIM; Ticarcillin/Clavulanic Acid: PIP; Piperacillin; PTZ; Piperacillin/Tazobactam: CTX; Cefotaxime: PM; Cefepime; IPM; Imipenem: MEM; Meropenam: AK; Amikacin: CN; Gentamicin; TOB; Tobramycin: CIP; Ciprofloxacin: Min; Minocycline: CT; Colistin: SXT; Trimethoprim/Sulfamethxazole. Interpretations breakpoint of antibiotic susceptibility is based on CLSI criteria.

**Figure 3 antibiotics-10-01062-f003:**
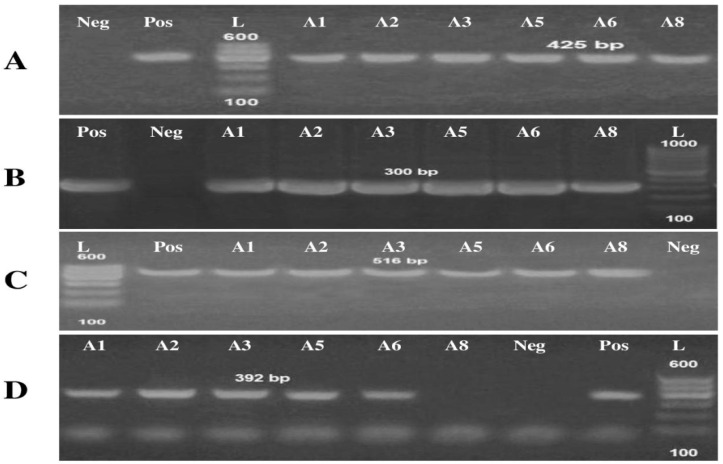
1.5% agarose gel electrophoresis of multiplex PCR of virulence and antibiotic resistant genes for the *A. baumannii* isolates. (**A**): rec*A* (425 bp) gene for identification *A. baumannii*. Lane L: Gel pilot 100 bp plus ladder (cat.no. 239045) supplied from QIAGEN (USA) as molecular size DNA marker Lane Pos: Positive control for rec*A* gene confirmed by reference laboratory for quality control. Lane Neg: Negative control. A1 for urine isolates; A5 for wound isolates, A2, A3, A6, A8 for respiratory isolates. (**B**): iutA (300 bp) virulence gene. (**C**,**D**): bla_TEM_ (516 bp) and bla_SHV_ (392 bp) antibiotic resistance genes, respectively.

**Figure 4 antibiotics-10-01062-f004:**
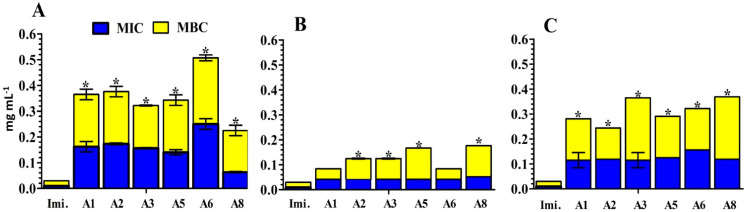
The differences of MIC and MBC values of Imipenem and *S. aromaticum* different extracts against *A. baumannii* isolates. Shown are the medians from at least three independent measurements for MIC (blue) and MBC (yellow). *S. aromaticum* different extracts (100 mg mL^−1^) (**A**): aqueous extract; (**B**): ethanolic extract; and (**C**): ethyl acetate extract. Imi: Imipenem (10 mg mL^−1^). The error bars indicate the interquartile range. Significant differences between the data sets are marked by asterisks (*p* < 0.05; Kruskal–Wallis test and post hoc Dunn’s multiple comparisons). A1: A8 are the *A. baumannii* isolates codes. A1: isolates from urine sample; A5: from wound samples; A2, A3, A6, and A8 for respiratory sample.

**Figure 5 antibiotics-10-01062-f005:**
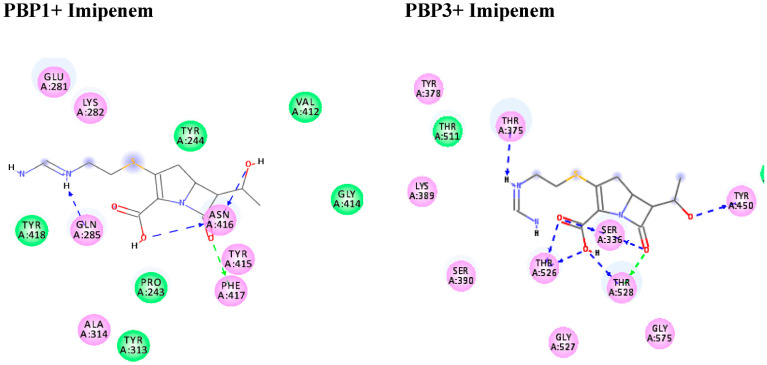
Molecular interactions of Imipenem with penicillin-binding protein 1 and 3 (PBP1 and PBP3) in *Acinetobacter baumannii*. Shown are the 2D binding modes upon docking. HBs are represented in blue and green dotted line colors.

**Figure 6 antibiotics-10-01062-f006:**
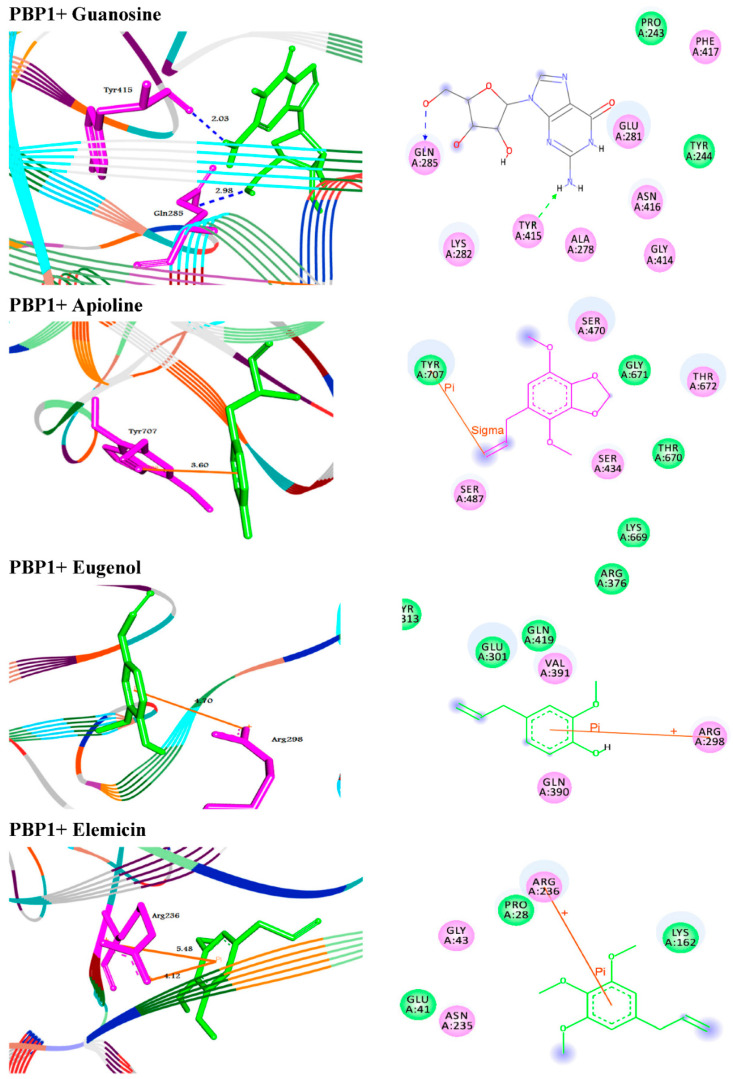
Molecular interactions of Guanosine, Apioline, Eugenol, and Elemicin with penicillin-binding protein 1 (PBP1) in *Acinetobacter baumannii*. Shown are the 3D (**left**) and 2D (**right**) binding modes upon docking. HBs are represented in blue and green dotted line colors while π- interactions are shown in yellow.

**Figure 7 antibiotics-10-01062-f007:**
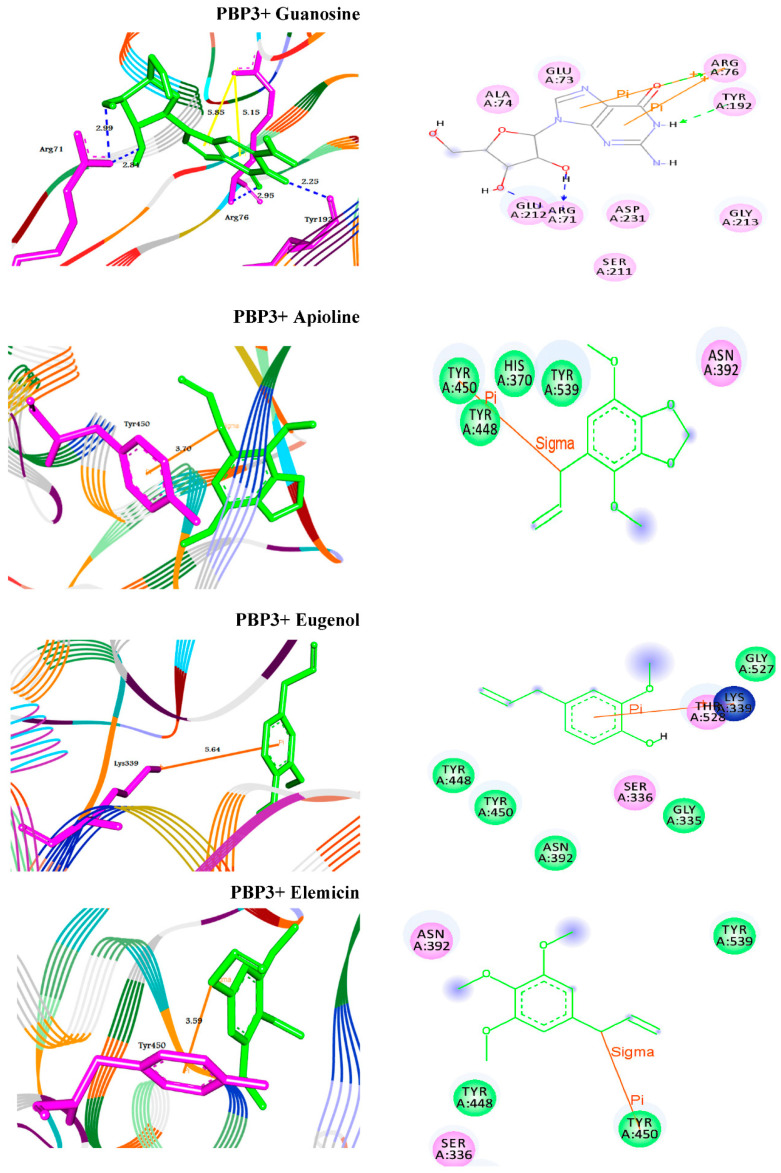
Molecular interactions of Guanosine, Apioline, Eugenol, and Elemicin with penicillin-binding protein 3 (PBP3) in *Acinetobacter baumannii*. Shown are the 3D (**left**) and 2D (**right**) binding modes upon docking. HBs are represented in blue and green dotted line colors while π- interactions are shown in yellow and orange line colors.

**Table 1 antibiotics-10-01062-t001:** Data and patient characters for the examined clinical samples.

Female		Male		Patient Character	
56.88%	No = 63	43.12%	No = 47		
28.5	18	12.77	6	below (30)	Age (year)
60.3	38	38.30	18	between (30–60)	
11.2	7	48.93	23	above (60)	
Prevalence of underlying disease					
20.63	9	6.38	3	Respiratory disease.	
16.1	7	10.63	5	Diabetes mellitus.	
27.0	17	21.3	10	Diabetic and hypertension	
15.87	10	10.63	5	Diabetic and respiratory disease	
11.11	7	17.02	8	Diabetic (respiratory disease and hypertension)	
9.51	6	10.63	5	Patients with elevated liver function test.	
11.1	7	19.16	9	Patients with elevated kidney function test.	
0.0	0	4.26	2	Catheter presence	

**Table 2 antibiotics-10-01062-t002:** Virulence and antibiotic resistant genes for the *Acinetobacter baumannii* isolates.

Types of Samples (Isolate No.)	Virulence Genes		Antibiotic Resistance Genes	
	*rec*A	*iut*A	bla_TEM_	Bla_SHV_
Urine (A1)	1	1	1	1
Wound (A5)	1	1	1	1
Sputum (A2,A3,A6, A8 *)	4	4	4	3
Total	6	6	6	5

A1: isolates from urine sample; A5: from wound samples; A2, A3, A6 and A8 for respiratory sample. * A8 isolate missed Bla_SHV_.

**Table 3 antibiotics-10-01062-t003:** GC-MS for bioactive chemical components of different extracts from *S. aromaticum* *.

Aqueous Extracts					
No.	RT (min)	Compound Name	M. Formula	M.wt	Area (%)
1	7.38	Limonene	C_10_H_16_	136	1.56
2	8.61	α-Pinene	C_10_H_16_	136	18.82
3	11.26	(2E)-3,7-dimethylocta-2,6-dienal	C_10_H_16_O	152	0.86
4	13.18	Camphor	C_10_H_16_O	152	11.75
5	14.33	Cyclododecene	C_12_H_22_	166	0.76
6	14.80	2,4 Decadienal	C_10_H_16_O	152	1.05
7	18.06	α -Chamigrene	C_15_H_24_	204	0.52
8	18.21	à-Guaiene	C_15_H_24_	204	0.59
9	18.57	Beta-caryophyllene	C_15_H_24_	204	15.12
10	19.32	Ethyl benzoylacetate	C_11_H_12_O_3_	192	1.05
11	19.52	Globulol	C_15_H_26_O	222	11.35
12	22.76	Apioline	C_12_H_14_O_4_	222	5.45
13	29.05	Hexadecanoic acid	C_16_H_32_O_2_	256	4.59
14	29.78	Loganetin	C_11_H_16_O_5_	228	8.51
15	30.75	Isobergapten	C_12_H_8_O_4_	216	1.09
16	32.66	Oleic acid	C_18_H_34_O_2_	282	14.52
17	32.86	Isochiapin B	C_20_H_26_N_2_O_2_	326	0.62
18	42.01	Lucenin	C_27_H_30_O_16_	610	0.56
**Ethanolic extract**					
1	28.33	Pentenenitrile	C_5_H_7_N	81	4.37
2	29.66	Ethyl oleate	C_20_H_38_O_2_	310	2.54
3	32.17	cis-10-Nonadecenoic acid	C_19_H_36_O_2_	296	5.37
4	35.12	Indole	C_8_H_7_N	117	6.83
5	37.21	Guanosine	C_10_H_13_N_5_O_5_	283	8.91
6	39.55	Oleic acid	C_18_H_34_O_2_	282	27.22
7	39.75	Chlorozotocin	C_9_H_16_ClN_3_O_7_	313	0.52
8	40.62	1-Eicosene	C_20_H_40_	280	6.30
9	46.77	Nonadecene	C_19_H_38_	266	1.20
10	48.85	3-Hexacosanol	C_26_H_54_O	382	2.21
11	51.96	Nonacosane	C_29_H_60_	408	0.36
12	53.70	Dodecanoic acid	C_12_H_24_O_2_	200	2.34
**Ethyl acetate extract**					
1	13.23	Linalool	C_10_H_18_O	154	1.24
2	13.94	Carveol	C_10_H_16_O	152	8.08
3	14.46	Citral	C_10_H_16_O	152	13.48
4	15.07	Eugenol	C_10_H_12_O_2_	164	1.91
5	15.23	1-Hexadecene	C_16_H_32_	224	0.16
6	18.30	Phenol, 2,4-bis(1,1-dimethylethyl)-	C_14_H_22_O	206	0.13
7	20.62	Elemicin	C_12_H_16_O_3_	208	3.32
8	21.73	Farnesyl acetate	C_17_H_28_O_2_	264	1.44
9	22.42	Heptadecane	C_17_H_36_	240	0.28
10	23.94	Tetradecanoic acid	C_14_H_28_O_2_	228	0.73
11	27.11	Hexadecanoic acid, methyl ester	C_17_H_34_O_2_	270	0.12
12	27.71	Alantolactone	C_15_H_20_O_2_	232	6.36
13	28.16	Hexadecanoic acid	C_16_H_32_O_2_	256	11.95
14	28.56	Eremanthin	C_15_H_18_O_2_	230	2.76
15	30.32	Linoleic acid methyl ester	C_19_H_34_O_2_	294	0.73
16	30.43	Oleic acid methyl ester	C_19_H_36_O_2_	296	0.37
17	31.60	Linoleic acid	C_18_H_32_O_2_	280	36.16
18	31.71	Oleic Acid	C_18_H_34_O_2_	282	6.39
19	31.95	Octadecanoic acid	C_18_H_36_O_2_	284	3.23
20	43.71	Ethyl iso-allocholate	C_26_H_44_O_5_	436	0.22

*: RT: Retention time per minute; active compounds detected by GC mass; area (%): percentage of compound; M. formula: molecular formula; M.wt: molecular weight of the compound (g/mol).

**Table 4 antibiotics-10-01062-t004:** The binding affinity (kcal/mol) of some phyto-compounds and Imipenem with PBP1 and PBP3 after molecular docking.

	PBP1 (3udx)			PBP3 (3ue3)		
	Binding Energy kcal/mol	Docked Complex (Amino Acid–Ligand)	Distance (Å)	Binding Energy kcal/mol	Docked Complex (Amino Acid–Ligand)	Distance (Å)
Imipenem	−6.8	H–bond		−6.5	H–bond	
		GLN285:NE2–Imipenem	2.15		SER336–Imipenem	1.94
		ASN416–Imipenem	2.55		SER336–Imipenem	2.17
		PHE417–Imipenem	2.96		TYR450–Imipenem	2.23
					THR526–Imipenem	2.18
					THR526–Imipenem	1.99
					THR528–Imipenem	2.53
					THR528–Imipenem	2.97
Guanosine	−7.1	H–bond		−8.1	H–bond	
		GLN285:NE2–Guanosine	2.98		ARG71:NH1–Guanosine	2.97
		TYR415:O–Guanosine			ARG71:NH1–Guanosine	2.84
			2.03		ARG76:N–Guanosine	2.93
					TYR192:O–Guanosine	2.25
					π–π interaction	
					ARG76:N–Guanosine	4.14
					ARG76:N–Guanosine	5.06
Apioline	−5.6	π–sigma interaction		−6.0	π–sigma interaction	
		TYR707–Apioline	3.60		TYR450–Apioline	3.70
Eugenol	−5.4	π–cation interaction		−5.8	π–cation interaction	
		ARG298:NH2–Eugenol	4.70		LYS339:NZ–Eugenol	5.64
Elemicin	−5.3	π–cation interaction		−5.2	π–sigma interaction	
		ARG236:NH1–Elemicin	5.48		TYR450–Elemicin	3.59
		ARG236:NH2–Elemicin	4.12			

The 3D structures of PBP1 and PBP3 were downloaded from the protein data bank (with pdb IDs: 3udx and 3ue3, respectively).

**Table 5 antibiotics-10-01062-t005:** Primers sequences, target genes, amplicon sizes, and cycling conditions.

Target Gene	Sequence	Amplified Segment (bp)	Primary Denaturation	Amplification (35 Cycles)				References
				Secondary Denaturation	Annealing	Extension	Final Extension	
**Virulence genes used for *Acinetobacter baumannii* isolates ***								
*recA*	CCTGAATCTTCTGGTAAAAC GTTTCTGGGCTGCCAAACATTAC	425	94 °C/5 min	94 °C/30 s	50 °C/45 s	72 °C/30 s	72 °C/10 min	[53]
*iut*A	GGCTGGACATGGGAACTGG CGTCGGGAACGGGTAGAATCG	300	94 °C/5 min	94 °C/30 s	63 °C/30 s	72 °C/45 s	72 °C/7 min	[54]
**Antibiotics resistance genes**								
*bla_TEM_*	ATCAGCAATAAACCAGC CCCCGAAGAACGTTTTC	516	94 °C/5 min	94 °C/30 s	54 °C/40 s	72 °C/45 s	72 °C/10 min	[55]
*bla_SHV_*	AGGATTGACTGCCTTTTTG ATTTGCTGATTTCGCTCG	392	94 °C/5 min	94 °C/30 s	54 °C/40 s	72 °C/45 s	72 °C/10 min	[55]

*: The specific sequences that were amplified for each of the used primers (Metabion, Germany).

## Data Availability

The data presented in this study are available on request from the corresponding author. The data are not publicly available due to privacy restrictions.

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
