# Peer review of "Syzygium aromaticum Extracts as a Potential Antibacterial Inhibitors against Clinical Isolates of Acinetobacter baumannii: An In-Silico-Supported In-Vitro Study"

_antibiotics, 2021, doi:10.3390/antibiotics10091062_

Round 1
Reviewer 1 Report
Line 35: Perhaps remove the "and".
Line 36: "coccobacilli, common in hospital settings, particularly in intensive care"
Line 37: "skin and wound diseases" might not be the best wording.
Line 40: "inactivation of enzymes and formation of biofilm"
Line 42: Remove the semicolon.
Line 45: "60% of all antimicrobial agents". Are you referring to antibiotics only?
Line 54: Confusing, please rewrite.
Line 58 to 60: The end of this sentence is a bit confusing, maybe try rewriting.
Line 59: Remove the comma.
Overall, the introduction could be improved. Some of the wording is not the most appropriate and it could be shorter and clearer.
Line 85: "...were above 60 years old, while 21.8% (18 female; 6 male) were below 30 years old."
The quality of the images is not the best. Additionally, in the title of Figure 2, you should replace ug/ml for μg/mL.
Line 117: Replace "While" with "Around".
Line 118: "...and 70% are resistant"
Line 118 to 119: Please rewrite or connect the sentence with the previous one, as it is not clearly written.
Line 124 and 125: Which ones were considered for further experiments? All mentioned. Please clarify.
Line 145: Please remove the "some" as it is ambiguous.
Line 164 to 166: Very confusing, please clarify.
Line 168: GC-MS is sufficient.
Line 172: Be careful with the italicization of words. In this case, since the title is in italics, "In vitro" and "S. aromaticum" should not be italicized.
Line 176 to 177: What were the exposure times assayed?
Line 178: "varying from" instead of "varied from"
Line 179 to 182: Please rewrite, this part is very confusing. Perhaps being more concise can improve this.
Figure 5: Again, quality can be improved. The lettering inside the small circles is almost imperceptible.
Line 218: Remove "The".
Line 221: Remove "the".
Line 223: This sentence should be reformulated. For example: "...that upset the kidney, leading to pyelonephritis, as well as the bladder, resulting in cystitis."
Line 224: Remove "Though" and try connecting this sentence with the one that follows.
Line 229 to 232: This seems a bit redundant.
Line 232: "has become a major concern".
Line 239: "While 70-80%" it is not necessary to be a new sentence, please connect it with the previous one.
Line 240: What do you mean by "become sensitive to Colistin"?
Line 252 to 254: Again, seems redundant as it has been explained in the introduction.
Line 257: Remove "is".
Line 264: "that results in". Also, please connect with previous sentence.
Line 268: Participate in what?
Line 269: "alters" instead of "altering"; "causes" instead of "causing"
Line 276: Please italicise in vitro and in vivo.
Line 290: Which molecule displayed what?
Line 379: There is a "m" missing in "45 μm membrane filter". Also, have you considered using a 0.22 μm filter?
Line 384: Why did you use an OD of 595 to adjust the inoculum?
Line 390: You used p-iodonitrotetrazolium violet chloride to assess MIC. Doesn't this dye evaluate metabolic inhibition as it is reduced by the bacteria? How can you verify inhibitory activity this way?
Line 437 to 438: Confusing, please rewrite.
Author Response
Response to Reviewer 1 Comments
Thank you very much for handling the review process of our manuscript ID: antibiotics-1323583 Entitled " Syzygium aromaticum Extracts as a Potential Antibacterial Inhibitors against Clinical Isolates of Acinetobacter baumannii: An In-Silico-Supported In-Vitro Study ".
We thankfully acknowledge your positive comments on our manuscript.
We greatly appreciate your time and consideration, and we hope that you would find our revised manuscript suitable now for publication.
Point 1: Line 35: Perhaps remove the "and".
Response 1: Corrected
Point 2: Line 36: "coccobacilli, common in hospital settings, particularly in intensive care"
Response 2: The above point was corrected carefully
Point 3: Line 37: "skin and wound diseases" might not be the best wording.
Response 3: The meaning of these words have been edited and clarified with other expressions
Point 4: Line 40: "inactivation of enzymes and formation of biofilm"
Response 4: Corrected
Point 5: Line 42: Remove the semicolon.
Response 5: Semicolon has been removed.
Point 6: Line 45: "60% of all antimicrobial agents". Are you referring to antibiotics only?
Response 6: Yes, and it was corrected accordingly.
Point 7: Line 54: Confusing, please rewrite.
Response 7: The above line was checked and clarified.
Point 8: Line 58 to 60: The end of this sentence is a bit confusing, maybe try rewriting.
Response 8: The above lines were checked and edited according to the reviewer comment.
Point 9: Line 59: Remove the comma.
Response 9: The comma has been removed.
- Overall, the introduction could be improved. Some of the wording is not the most appropriate and it could be shorter and clearer.
The introduction part has been checked again very carefully, summarized and clarified.
Point 10: Line 85: "...were above 60 years old, while 21.8% (18 female; 6 male) were below 30 years old."
Response 10: The above line was corrected.
Point 11: The quality of the images is not the best. Additionally, in the title of Figure 2, you should replace ug/ml for μg/mL.
Response 11: The quality of the figure has been improved with a high resolution one, and the title of the figure 2 has been corrected as well.
Point 12: Line 117: Replace "While" with "Around".
Response 12: corrected
Point 13: Line 118: "...and 70% are resistant"
Response 13: corrected
Point 14: Line 118 to 119: Please rewrite or connect the sentence with the previous one, as it is not clearly written.
Response 14: The meaning of these sentences has been edited and clarified according to the reviewer comment.
Point 15: Line 124 and 125: Which ones were considered for further experiments? All mentioned. Please clarify.
Response 15: corrected and clarified carefully.
Point 16: Line 145: Please remove the "some" as it is ambiguous.
Response 16: removed
Point 17: Line 164 to 166: Very confusing, please clarify.
Response 17: The meaning of these sentences has been edited and clarified.
Point 18: Line 168: GC-MS is sufficient.
Response 18: corrected.
Point 19: Line 172: Be careful with the italicization of words. In this case, since the title is in italics, "In vitro" and "S. aromaticum" should not be italicized.
Response 19: this was considered and corrected.
Point 20: Line 176 to 177: What were the exposure times assayed?
Response 20: the exposure times have been mentioned.
Point 21: Line 178: "varying from" instead of "varied from"
Response 21: corrected.
Point 22: Line 179 to 182: Please rewrite, this part is very confusing. Perhaps being more concise can improve this.
Response 22: The whole sentences were improved.
Point 23: Figure 5: Again, quality can be improved. The lettering inside the small circles is almost imperceptible.
Response 23: The quality of figure 5 has been improved with a high resolution, and all the lettering inside the circles is now clear.
Point 24: Line 218: Remove "The".
Response 24: corrected
Point 25: Line 221: Remove "the".
Response 25: corrected
Point 26: Line 223: This sentence should be reformulated. For example: "...that upset the kidney, leading to pyelonephritis, as well as the bladder, resulting in cystitis."
Response 26: The sentence was reformulated according to the reviewer comment.
Point 27: Line 224: Remove "Though" and try connecting this sentence with the one that follows.
Response 27: corrected
Point 28: Line 229 to 232: This seems a bit redundant.
Response 28: the whole sentence was deleted accordingly.
Point 29: Line 232: "has become a major concern".
Response 29: corrected
Point 30: Line 239: "While 70-80%" it is not necessary to be a new sentence, please connect it with the previous one.
Response 30: corrected
Point 31: Line 240: What do you mean by "become sensitive to Colistin"?
Response 31: corrected and clarified.
Point 32: Line 252 to 254: Again, seems redundant as it has been explained in the introduction.
Response 32: the whole sentence was deleted accordingly.
Point 33: Line 257: Remove "is".
Response 33: corrected
Point 34: Line 264: "that results in". Also, please connect with previous sentence.
Response 34: corrected
Point 35: Line 268: Participate in what?
Response 35: corrected
Point 36: Line 269: "alters" instead of "altering"; "causes" instead of "causing"
Response 36: corrected
Point 37: Line 276: Please italicise in vitro and in vivo.
Response 37: corrected
Point 38: Line 290: Which molecule displayed what?
Response 38: corrected and clarified
Point 39: Line 379: There is a "m" missing in "45 μm membrane filter". Also, have you considered using a 0.22 μm filter?
Response 39: the missing letter was added- only 45 μm membrane filter was used in our experiments.
Point 40: Line 384: Why did you use an OD of 595 to adjust the inoculum?
Response 40: Using OD595 to adjust the bacterial inoculum in order to start the initial with approximately the same bacterial phase. Also we considered the OD595 to measure the bacterial viability in the plate reader.
Point 41: Line 390: You used p-iodonitrotetrazolium violet chloride to assess MIC. Doesn't this dye evaluate metabolic inhibition as it is reduced by the bacteria? How can you verify inhibitory activity this way?
Response 41: YES, it is a growth indicator that gives a red color (formazone- complex with the active bacterial enzymes in case of the bacteria are still live) and we can measure the colour intensity by a micro-plate reader (measure the OD) or otherwise by recording the inhibitory activity by observing the concentrations which the dye gives no color.
Point 42: Line 437 to 438: Confusing, please rewrite.
Response 42: corrected and clarified.
Reviewer 2 Report
Mahmoud et al. describe the antibacterial activity of S. aromaticum extracts against A. baumannii.
Major revisions:
- As ethanol itself has antibacterial properties, control samples are needed for ethanol.
- Why only guanosine is selected for the docking study but not alpha-pinene, oleic acid and/or other highly presented compounds?
Minor revisions:
- colon, semicolon and brackets are used incorrectly;
- give the pdb codes of PBP1a and PBP3.
Author Response
Response to Reviewer 2 Comments
Thank you very much for handling the review process of our manuscript ID: antibiotics-1323583 Entitled " Syzygium aromaticum Extracts as a Potential Antibacterial Inhibitors against Clinical Isolates of Acinetobacter baumannii: An In-Silico-Supported In-Vitro Study".
We thankfully acknowledge your positive comments on our manuscript.
We greatly appreciate your time and consideration, and we hope that you would find our revised manuscript suitable now for publication.
Reviewer 2 Comments
Mahmoud et al. describe the antibacterial activity of S. aromaticum extracts against A. baumannii.
Major revisions:
Point 1: As ethanol itself has antibacterial properties, control samples are needed for ethanol.
Response 1: As it was written in the text (Materials & Methods part) that the obtained extracts were filtered through a buchner funnel with Whatman No.1 filter paper and evaporated by a rotary evaporator (BUCHI R-114, Switzerland) under reduced pressure to dryness at 45°C. All extracts- residue were dissolved in dimethyl sulfoxide (DMSO).
We just used the residue after a complete evaporation of ethanol.
Point 2: Why only guanosine is selected for the docking study but not alpha-pinene, oleic acid and/or other highly presented compounds?
Response 2: Given the great importance of pyrimidine compounds, especially in recent years, which are represented in wide applications in many fields, especially in the medical aspect, as antibacterial1 agents. In addition, pyrimidines are nitrogen heterocyclic aromatic compounds with great interest as they constitute an important class of natural products. The substituted pyrimidines such as Brodiprim, Iclaprim, Trimethoprim and Pyrimethamine (as shown in the following Figure) are biologically important compounds and act as effective antibacterial drugs2. Therefore, Guanosine with pyrimidine ring was selected as drug candidate against PBP1 and PBP3 proteins.
Figure. Examples of antibacterial agents containing pyrimidine ring.
- Horchani, M. et al. New pyrazolo-triazolo-pyrimidine derivatives as antibacterial agents: Design and synthesis, molecular docking and DFT studies. J. Mol. Struct. 1199, 127007 (2020).
- Sharma, V., Chitranshi, N. & Agarwal, A. K. Significance and Biological Importance of Pyrimidine in the Microbial World. Int. J. Med. Chem. 2014, 1–31 (2014).
Minor revisions:
Point 3: colon, semicolon and brackets are used incorrectly;
Response 3: colon, semicolon and brackets were checked along the text and corrected according.
Point 4: give the pdb codes of PBP1a and PBP3.
Response 4: The pdb coded of PBP1 and PBP3 has been added in Table 5 and in the text as well.

Reviewer 3 Report
Eleasawy and coworkers present an experimental study complemented by molecular docking, concerning the antibacterial activity of Syzygium aromaticum extracts and Imipenem derivatives against A. baumannii. The study is exhaustive and provides valuable information for teams devoted to the development of novel agents against A. baumannii. For these reasons we consider that this work can be published in Antibiotics with minor revisions.
The docking results should be further exploited, even if it is a first approximation to the guanosine-A. baumannii interactions. The authors should mention the main fragments of the candidates interacting with the target. In addition, the quality of figure 5 should be improved.
In the abstract, it reads:
“Imipenem is the most efficient antibiotic against Acinetobacter baumannii infection, but new research has shown that the organism has also evolved resistance to imipenem.”
It should read:
“Imipenem is the most efficient antibiotic against Acinetobacter baumannii infection, but new research has shown that the organism has also developed resistance to this agent.”
In line 75, the authors state:
"Docking and dynamic studies were conducted to determine and compare the interactions of both Syzygium aromaticum compounds and the antibiotic imipenem ..."
There is no dynamic study in this work
In line 248, the authors state:
"This result is in agreement with the result reported in turkey ..."
The authors’ names should be mentioned, and not their country
ml should be replaced by mL along the text
Author Response
Response to Reviewer 3 Comments
Thank you very much for handling the review process of our manuscript ID: antibiotics-1323583 Entitled " Syzygium aromaticum Extracts as a Potential Antibacterial Inhibitors against Clinical Isolates of Acinetobacter baumannii: An In-Silico-Supported In-Vitro Study ".
We thankfully acknowledge your positive comments on our manuscript.
We greatly appreciate your time and consideration, and we hope that you would find our revised manuscript suitable now for publication.
Reviewer 3 Comments
Eleasawy and coworkers present an experimental study complemented by molecular docking, concerning the antibacterial activity of Syzygium aromaticum extracts and Imipenem derivatives against A. baumannii. The study is exhaustive and provides valuable information for teams devoted to the development of novel agents against A. baumannii. For these reasons we consider that this work can be published in Antibiotics with minor revisions.
Point 1: The docking results should be further exploited, even if it is a first approximation to the guanosine-A. baumannii interactions. The authors should mention the main fragments of the candidates interacting with the target. In addition, the quality of figure 5 should be improved.
Response 1: The docking results have been further exploited, more explanations have been added and the main fragments of the candidates interacting with the target have been clarifies as well.
Point 2: In the abstract, it reads:
“Imipenem is the most efficient antibiotic against Acinetobacter baumannii infection, but new research has shown that the organism has also evolved resistance to imipenem.”
It should read:
“Imipenem is the most efficient antibiotic against Acinetobacter baumannii infection, but new research has shown that the organism has also developed resistance to this agent.”
Response 2: the sentence has been corrected according to the reviewer comments.
Point 3: In line 75, the authors state:
"Docking and dynamic studies were conducted to determine and compare the interactions of both Syzygium aromaticum compounds and the antibiotic imipenem ..."
There is no dynamic study in this work
Response 3: corrected
Point 4: In line 248, the authors state:
"This result is in agreement with the result reported in turkey ..."
The authors’ names should be mentioned, and not their country
Response 4: Corrected
Point 5: ml should be replaced by mL along the text
Response 5: ml has been changed to mL along all the manuscript.
Round 2
Reviewer 2 Report
Corrections are not included in the text.
Author Response
All the corrections are now included in the manuscript. such as phyto-compounds (Apioline, Eugenol, and Elemicin) were successfully docked to PBP1 and PBP3, and it was indexed in Table 5, Figur 6 and figure 7
